

# Valence moderates the effect of stimulus-hand proximity on conflict processing and gaze-cueing

Sven Hoffmann[1,2], Rico Fischer[3] and Roman Liepelt[1]

[1] Department of General Psychology, Judgement, Decision Making, Action, University of Hagen, Hagen, Germany
[2] Performance Psychology, Institute of Psychology, German Sport University Cologne, Cologne, Germany
[3] Department of General Psychology, Institute of Psychology, University of Greifswald, Greifswald, Germany

Corresponding author
Sven Hoffmann,
sven.hoffmann@fernuni-hagen.de

## ABSTRACT

An effective interaction with the environment requires adaptation of one's own behaviour to environmental demands. We do so by using cues from our environment and relating these cues to our body to predict the outcomes of events. The recent literature on embodied cognition suggests that task-relevant stimuli, presented near the hands, receive more attentional capacity and are processed differently than stimuli, presented spatially more distant to our body. It has also been proposed that near-hand processing is beneficial to conflict resolution. In the current study, we tested the assumption of an attentional bias towards the near hand space in the context of our previous work by combining a cueing paradigm (allocation of visual attention) with a conflict processing paradigm (Simon task) in the near *vs* far hand space. In addition, the relevance of processing was manipulated by using affective (angry *vs* neutral smileys) gaze cues (*i.e.*, varying the valence of the cues). Our results indicate that (a) the interaction of valence × cue congruency × hand proximity was significant, indicating that the cueing effect was larger for negative valence in the proximal condition. (b) The interaction of valence × Simon compatibility × stimulus-hand proximity interaction was significant, indicating that for negative valence processing, the Simon effect was smaller in the proximal than in the distal stimulus-hand condition. This effect was at least numerically but not significantly reversed in the neutral valence condition. (c) Overall, cue congruency, indicating the correct *vs* incorrect attention allocation to the target stimulus onset, did not reveal any effect on Simon compatibility × stimulus-hand proximity. Our results suggest that valence, the allocation of attention, and conflict, seem to be decisive factors determining the direction and strength of hand proximity effects.

# INTRODUCTION

All day, we monitor, adapt, and evaluate our actions concerning environmental demands. To do so, we use environmental cues and relate them to our bodily state to predict the

outcomes of events. In this regard, the distinction between personal and extra-personal space as well as the salience of the events are core aspects. These dimensions must be integrated to weight and process relevant internal and/or external information concerning action.

A phenomenon reflecting this weighting is the hand proximity effect. It can be observed, that if visual stimuli are presented near the hands, they are processed faster and more efficient, compared to stimuli presented in a more distant position (*Hari & Jousmäki, 1996*). On a more complex level, task-relevant stimuli that are presented near the hands receive more attentional capacity and are processed differently than stimuli in a spatially more distant position: for instance, attentional selection seems to be biased in near hands conditions (*Reed, Grubb & Steele, 2006*) and an increased visual working memory capacity can be found in this condition (*Tseng & Bridgeman, 2011*). It has been suggested that these effects reflect the fact that objects near the hands are interpreted as being relevant for action, or, in case of obstacles, should be avoided. Thus, stimuli presented in near hand space should catch more attention (*Brockmole et al., 2013*). Indeed, higher performance, in terms of faster response times and higher accuracy, have been found in simple discriminatory tasks for stimuli presented near the hands (*Whiteley et al., 2004*). Further, it has been hypothesized that objects presented near the hands benefit from improved visual processing since these objects appear to be potential candidates for following actions (*Abrams et al., 2008*). Thus, one might conclude that a hand acting on a close-by object alters the way the object is processed (*Liepelt, 2014*).

## The effect of task demands for objects presented near the hands

The stimulus processing with varying hand positions has been investigated in a wide variety of tasks. However, the underlying cognitive mechanisms of stimulus processing in this context are still not fully understood. It has been shown that hand proximity effects depend on task demands (*Liepelt & Fischer, 2016*), and can even be reversed depending on the level and complexity of processing required in the respective task. It was shown that the Simon effect[1] was reduced for the hands-close condition if the task was about higher-level numerical judgements but increased if the Simon task was about lower-level perceptual feature discrimination. Furthermore, it has been suggested that there is an enhancement in spatial processing for stimuli appearing in the area near the hands (*Reed, Grubb & Steele, 2006*; *Abrams et al., 2008*). Further evidence that could be interpreted in terms of increased spatial feature processing comes from a visual attention paradigm: the near-hand position leads to a delayed or interrupted decoupling of attention from the spatially targeted object (*Abrams et al., 2008*). These findings indicate that the attentive processing of task-relevant and task-irrelevant information can be modified when the stimuli are presented near the hands.

Further empirical observations suggest increased cognitive control during stimulus processing near the hands reflected in diminished interference effects in a non-spatial Stroop test (*Davoli et al., 2010*), reduced task-switching costs for stimuli that appeared near the hands (*Weidler & Abrams, 2014*), and reduced dual-task crosstalk (*Fischer & Liepelt, 2020*). The reduced interference effects for the stimulus-proximate hand position suggest

[1] The Simon effect occurs when participants are asked to respond to a stimulus that appears on one side of a screen (*e.g.*, a red square on the left side), but the response requires pressing a button on the opposite side (*e.g.*, the right button). Despite the fact that the location of the stimulus is irrelevant to the task, participants typically respond faster and more accurately when the stimulus appears on the same side as the response hand (*e.g.*, when a red square appears on the right side and the right button needs to be pressed) compared to when the stimulus appears on the opposite side of the response hand (*e.g.*, when a red square appears on the left side and the right button needs to be pressed).

that stimuli near the hands receive increased levels of cognitive control in cognitively demanding tasks (*Weidler & Abrams, 2014*; *Liepelt & Fischer, 2016*; *Fischer & Liepelt, 2020*). For instance, a hand near condition reduces the amount of between task interference (*i.e.*, crosstalk) as compared to a hands far condition in a dual-task paradigm (*Fischer & Liepelt, 2020*). This finding provides evidence for the optimization of task shielding, a cognitive control process that supports the separation of two simultaneously processed tasks by protecting the processing of a prioritized task from the interference of a concurrently running secondary task. In addition to the typical impact of attention on the visual field near the reaction site, there is also evidence for an effect of task expectations, *i. e.*, a top-down influence (*Fischer & Liepelt, 2020*). The influence of both, bottom-up, like allocation of attention, (*Reed, Grubb & Steele, 2006*) and top-down information (*Fischer & Liepelt, 2020*) demonstrate the complexity of differential processing of stimuli presented near the hands.

The findings of a significantly smaller Simon effect in a relatively difficult Simon categorization task for the hands-near position can be attributed to increased cognitive control (*Weidler & Abrams, 2014*; *Fischer & Liepelt, 2020*). Thus, task difficulty and target-relevant parameters seem to be decisive factors for the direction and the strength of the hand proximity effects. This is supported by studies showing that hand proximity effects may in general be rather weak and sometimes even non-existing in tasks addressing basic processing of stimuli (*Andringa et al., 2018*; *Dosso & Kingstone, 2018*). *Andringa et al. (2018)* tested the effect with basic processing like visual search, tracing, and change detection and found absent or small effects, however, their main conclusion is that the near hands effect is sensitive to small variations in experiments. This is a rather unspecific notion as the used experiments tested rather diverse settings and not specific interactions with more complex material and tasks. The study by *Dosso & Kingstone (2018)* addressed basic visual processing and not higher order cognition, thus, this study cannot be generalized to more complex material as well. However, it seems like the hand proximity effect seems to be affected rather by higher order processes, compared to basic processes including visual attention.

In sum, the diversity of described phenomena regarding the hand proximity effect points to an involvement of attentional allocation and cognitive control. However, the interplay of attention allocation and cognitive control has not yet been addressed in detail. Further, there is a gap in research regarding a crucial dimension coding personal relevance of the task demands that is the interplay of emotional/affective valence and hand position.

## Attention and conflict as induced by gaze cueing and Simon effect

To continue this line of research, we focus on the allocation of attention *via* cueing and conflict as induced by the Simon task. Typically, in a cueing paradigm, a spatial cue indicates the position of the following target stimulus. This cue is only indicative at a certain probability for the target's position and can be incongruent with respect to the target's position, *i.e.*, pointing to the position contra-lateral to the target. A well-validated cueing paradigm is the gaze cueing paradigm (*Frischen, Bayliss & Tipper, 2007*). A typical finding in these experiments is a slowing of response times and a decrease of accuracy at

the incongruent spatial position (*Friesen & Kingstone, 1998*). One interpretation of this effect is that in the case of incongruent positions a shift in the allocation of spatial attention, away from the cued position, is necessary, leading to this response time increase. For example, eyes looking to the left will lead to faster left responses, compared to right responses (*Ansorge, 2003*). In addition, some studies found that the effect varies with the valence of the gaze cues while other studies did not (*Mathews et al., 2003*; *Tipples, 2006*; *Bonifacci et al., 2007*). The valence of a gaze cue, as well as its integration into one's prediction of target occurrence, seems to be relevant in this regard since the effect is modulated more strongly, if emotional processing is relevant to one's action (*Pecchinenda et al., 2008*). In a recent study it was found that cognitive control is involved in resolving interference from emotional information in a gaze cueing paradigm. More specifically, cognitive load enhanced gaze cueing effects for angry facial expressions and reduced the impact of gaze cueing when faces were neutral (*Pecchinenda & Petrucci, 2016*). Indeed, as soon as cognitive control is necessary to reduce interference, angry faces seem to bias cued spatial positions (*Holmes et al., 2013*). This is supported by the finding that gaze-orienting effects are modulated by valence like fear and anger, as reflected in early attention-related EEG potentials, *i.e.*, the P1 (*Lassalle & Itier, 2013*). Further, there seems to be an attentional bias with threatening stimuli at short stimulus onset asynchronies between cue and target (*Tipples, 2006*). In sum, it seems that the valence of the facial expression modulates the gaze cueing effect only in certain conditions.

Further, one has to consider the different processing levels of cues and facial expressions: the valence of the face seems to be processed faster compared to the direction of the gaze cue (*Fichtenholtz et al., 2009*). Also, it has been shown that objects, which are indicated by gaze cues are evaluated by taking into account their facial expression (*Bayliss et al., 2007*).

Here, we argue that one aspect that might be crucial concerning valence-dependent gaze cueing could be whether the facial expression of the cue is connected to the subjects' action. The hands-near effects might provide more insight here. In this study, we aimed to replicate the hands near effect observed in a semantic Simon task, but including a comparable lower-level mapping like the before mentioned studies. Also, we aim to extend studies manipulating the task relevance of the stimuli by adding potentially emotional aversive task cues. We combined a gaze-cueing paradigm to manipulate the allocation of visual attention with conflict processing (Simon task) in near *vs* far hand space. In addition, the relevance of processing was manipulated by using affective (angry *vs* neutral smileys) gaze cues.

### Hypotheses

Based on previous studies, we hypothesized that as soon as the gaze cue becomes action relevant, by altering the processing of the cue, this will alternate both, cueing and Simon effect. More specifically, since aversive cues capture more attention compared to neutral ones (*Bonifacci et al., 2007*), we hypothesize that as soon as the hand's position is close to a "threatful" cue, the cue becomes more relevant provoking a stronger gaze cueing and Simon effect. We assess whether the effect of valence is rather on attentional processes or

on conflict processing as both have been shown to play a role in the hand proximity effect. In other words, we assume that valence modulates (a) the interaction of cue and hand position (allocation of attention) as well as (b) the interaction of Simon and hand position (conflict processing).

## METHODS

### Participants

A total of 32 healthy subjects participated in the study, which is a sample size common on this research area. Data of one participant had to be excluded because of too high error rates (>40%). Exclusion criteria were self-reported somatic, neurological, or psychiatric illnesses. The inclusion criterion was age between 18 and 30 years. The remaining 31 participants (seven female, 24 male) were voluntary participating students at the German Sport University Cologne receiving course credits. The mean age was 23.06 years (range = 19–29 years, SD = 2.93). Handedness and vision of the subjects were documented using a questionnaire (22 right-handed, three left-handed, three ambidextrous). Twenty participants had no visual aid, five used glasses, and three had visual acuity correction using of contact lenses. Participants were informed in advance about the course of the experiment and agreed to participate by means of written informed consent. The present study was conducted according to the revision of the Helsinki Declaration of the *World Medical Association (2013)* and its ethics standards. The implementation of the study was covered by the ethics regulation of German Sport University Cologne. The data analysis was pseudo-anonymous coded, *i.e.*, personal data are not stored together with the participant's data (*e.g.*, names).

### Apparatus

The experiment took place in a laboratory of the institute. The room was sound attenuated, did not consist of windows and was lightly dimmed. The subjects were seated in front of a 24-inch LED monitor (1,920 × 1,080 pixels, 60 Hz, Model: 24GM77-B; LG Corporation, Seoul, South Korea) throughout the data acquisition. The Python package PsychoPy (version 1.82) was used for presentation of the visual stimuli and recording of behavioural data (*Peirce, 2008*). Behavioural data (response times and accuracy rates) were recorded using four computer mice (model: RX250; Logitech, Romanel-sur-Morges, Switzerland).

### General procedure

The experimenter informed the participants about the course of the experiment. All subjects agreed to informed consent. Subsequently, they completed a demographic questionnaire. Afterwards, the paradigm was instructed *via* the monitor, and participants had the opportunity to clarify questions and ambiguities. Participants had to conduct the paradigm block wise, using two alternating hand positions, near the body (hands far) or near the monitor (hands near). The order of those experimental blocks was fully balanced across participants. After completion of the first block, a 5-min break was provided during which the experimenter prepared the second block. After the break, the same paradigm was instructed, started, and performed again, but with another hand position. The order of

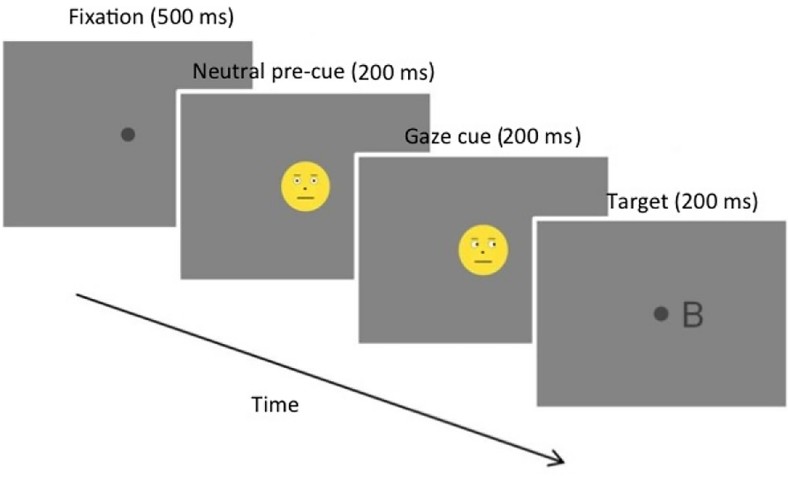

**Figure 1 Example trial of the gaze cueing paradigm.**

blocks was balanced between participants such that half of the participants started with the body condition and the other half started with the monitor condition. The measurement of one block took approx. 30 min. The total duration of the procedure was about 75 min.

## Experimental procedure

The conflict paradigm consisted of a combined gaze-cueing (attentional conflict) and Simon task (SR-conflict) with manipulation of hand position and emotional valence, *i.e.*, facial expression. We used schematic stimuli for gaze cues, *i.e.*, emoticons, as they are, beside basic emotional coding, more standardized compared to real faces. This is beneficial to use the paradigm *e.g.*, in EEG studies. Furthermore, a recent meta-analysis has shown that the gaze cuing effect is not moderated by face type (*e.g.*, real faces *vs* schematic faces), but that stimulus onset asynchrony (SOA) interacts to some degree with face type (*McKay et al., 2021*). They reported medium SOAs (401 to 600 ms) to be affected most by face type. Additionally, the strongest cueing effects have been observed for SOA < 300 ms (*Bayliss et al., 2007*). Also, the described meta-analysis has found evidence that the gaze cueing effect remains constant up to 600 ms and is decreasing with even longer SOAs (*McKay et al., 2021*). In sum, this allowed us to choose a short stimulus onset asynchrony (SOA) of 100 ms that is comparable to standard cueing paradigms. One trial is shown in Fig. 1 as an example. Each trial started with the presentation of a central fixation point. After 500–950 ms (randomized presentation duration), a neutral or angry looking emoticon was displayed centrally for a duration 200 ms. Then, the emoticon looked either to the left or to the right direction for 200 ms. Gazes could be congruent with respect to the following target (*i.e.*, same side), or incongruent (*i.e.*, opposite side). Afterwards, the emoticon was hidden and replaced by the target stimulus (letters). The left or right occurring target stimulus consisted of one of four letters. The appearance of either an F or a B indicated a left-hand button press with the left index finger, while a T or P indicated a click the on the right-hand button with the right index finger. We chose this regimen of mapping two letters to one finger, respectively, to increase task difficultly. The target could appear on the

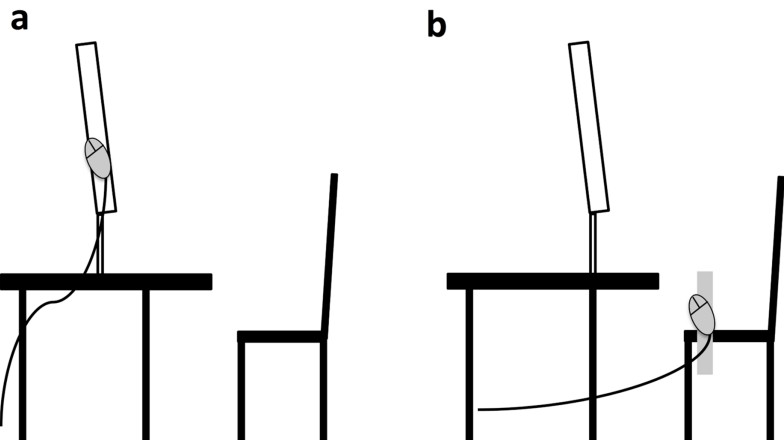

**Figure 2 Basic/schematic setup for the monitor (A) and body (B) condition.** Note that that the setup consisted of four computer mice attached to the left and right of the monitor/wooden device, respectively (c.g. main text).

side corresponding to the spatial position of the response button (Simon compatible) or on the other side (Simon incompatible). The target was presented for 200 ms. The response, with respect to the stimulus, had to be delivered within 850 ms, otherwise, the trial was considered invalid. Participants were instructed to respond as quickly and accurately as possible, and that the response with respect to the target should be made regardless of the expressed emotion of the smiley and the target's position. The viewing angle of the smileys was 1.5°, the viewing angle of the target was 0.5°. The distance (*i.e.*, viewing angle) between fixation point and the target stimulus (letter) was 2°. The dark grey target stimuli (RGB = [0.9, 0.9, 0.9]) were presented on a grey background (RGB = [0.1, 0.1, 0.1]). Note, that these are PsychoPy steering values/settings. Dependent variables were response times and accuracy rate.

The experiment consisted of 512 trials per hand position condition (close to monitor *vs* close to body) yielding a total of 1,024 trials. In the monitor condition, the hands were attached to the monitor (cf. Fig. 2). A mouse was attached to the right and left of the monitor, respectively. The arms were supported by arm cushions. At the body condition, the hands were on the thighs. Here, the mice were attached to a homemade wooden device. The mouse buttons had to be pressed with the index finger. The distance between the hands was 57 cm in both conditions, this was ensured by matching monitor width and width of the wooden device. The distance between the head and the monitor should be kept as constant as possible (about 60 cm) between the conditions. Due to different sizes of the participants this could not be realized perfectly, nevertheless it was tried to keep the distance between the conditions as constant as possible. Subjects were not allowed to keep their hands under the desk during body condition, as interference effects should be excluded that might occur if one does not see one's hands (*di Pellegrino & Frassinetti, 2000*).

## Data analysis

All analyses were made using GNU R (*R Core Team, 2022*). ANOVAs were conducted using the afex package (*Singmann et al., 2022*), contrast by the package emmeans (*Lenth, 2022*). All figures for made by using ggplot2 (*Wickham, 2016*), and tables were generated using the rempsyc package (*Thériault, 2022*). A complete list of used packages, the raw data, as well as the complete R markdown file can be downloaded from OSF (https://osf.io/sbxc8/). Response times were defined as the time between target onset and the first button press. For analysis of reaction time data only correct response times exceeding 100 ms after target onset and those being faster than two standard deviations above the mean RT were considered (RTs > 100 ms, RTs < mean RT + 2 SD) for each subject separately. Response times means were initially tested by an extended Shapiro-Wilks test to assess normality (*Royston, 1982*, *1983*). Subsequently, they were analysed by means of a four-way repeated measures ANOVA with the factors gaze cueing (incongruent *vs* congruent), Simon (compatible *vs* incompatible), hand position (body *vs* monitor), and valence (angry *vs* neutral). We report F-values, *p*-values (three decimals, except *p* < 0.001) and partial eta squared $\eta_p^2$. Due to df < 2 no violation of sphericity occurred. To increase statistical power regarding significance testing, we used additionally ANOVA permutation tests (10,000 permutations), as they are more robust regarding non-normality and when using small sample sizes. Further, it has been argued that they can handle the family wise error rate (FWER) in multiple comparisons (*Troendle, 1995*; *Maris & Oostenveld, 2007*; *Smith & Nichols, 2009*). Finally, we calculated Bayes factors for the initial four-way ANOVA (*Wagenmakers, 2007*; *Masson, 2011*; *Faulkenberry, 2018*), cf. R markdown file in the OSF-repository.

Significant interactions were followed by corresponding contrasts. They were corrected for Type I errors by the Benjamini & Yekutieli method (*Benjamini & Yekutieli, 2001*). Here, we report t-values, *p*-values, Cohen's d, and confidence intervals for Cohen's d.

Accuracy data, which is the number of erroneous responses within conditions, were transformed to percentages (=error rate). An extended Shapiro-Wilk test (*Royston, 1982*, *1983*) revealed that in most conditions, the accuracy data were not normally distributed, *Ws* < 0.9, *p* < 0.10. Thus, data were transformed by a square root transformation, which is an appropriate transformation for heavily right skewed data (*Tabachnick & Fidell, 2013*). These so transformed accuracy data were analysed by means of a repeated measures ANOVA with the factors gaze cueing (congruent, incongruent), Simon (compatible, incompatible), hand position (body, monitor), and valence (angry, neutral). Due to nominator *df* < 2 no violation of sphericity occurred. Thus, we report F-values, *p*-values (three decimals, except *p* < 0.001) and partial eta squared $\eta_p^2$. Again, we calculated a bootstrap ANOVA and provide Bayes factors. Significant interactions were followed by corresponding contrasts. They were corrected for Type I errors by the Benjamini & Yekutieli method (*Benjamini & Yekutieli, 2001*). We report t-values, *p*-values, Cohen's d, and confidence intervals for Cohen's d. In the tables, SE is the within-subjects standard error according to *Cousineau (2005, 2012)*. The confidence intervals (CI) refer to Cohen's

**Table 1  Four-way ANOVA for response time data.**

| Effect | SSn | dfn | SSd | dfd | MSEn | MSEd | F | $\eta_p^2$ | p | $p_{boot}$ | $BF_{10}$ |
|---|---|---|---|---|---|---|---|---|---|---|---|
| Cue | **2,741.04** | **1.00** | **7,420.33** | **30.00** | **2,741.04** | **247.34** | **11.08** | **0.27** | **0.002** | **0.00** | **23.47** |
| Simon | **83,824.00** | **1.00** | **42,954.38** | **30.00** | **83,824.00** | **1,431.81** | **58.54** | **0.66** | **<0.001** | **0.00** | **3,466,607.64** |
| Valence | **878.23** | **1.00** | **5,065.65** | **30.00** | **878.23** | **168.85** | **5.20** | **0.15** | **0.030** | **0.03** | **2.14** |
| Hand | 1,402.33 | 1.00 | 124,256.29 | 30.00 | 1,402.33 | 4,141.88 | 0.34 | 0.01 | 0.565 | 0.56 | 0.21 |
| Cue: Simon | **2,759.88** | **1.00** | **10,316.00** | **30.00** | **2,759.88** | **343.87** | **8.03** | **0.21** | **0.008** | **0.01** | **7.08** |
| Cue: Valence | 20.98 | 1.00 | 4,145.90 | 30.00 | 20.98 | 138.20 | 0.15 | 0.01 | 0.700 | 0.70 | 0.19 |
| Simon: Valence | 97.58 | 1.00 | 6,534.79 | 30.00 | 97.58 | 217.83 | 0.45 | 0.01 | 0.508 | 0.50 | 0.23 |
| Cue: Hand | 21.81 | 1.00 | 5,260.32 | 30.00 | 21.81 | 175.34 | 0.12 | 0.00 | 0.727 | 0.72 | 0.19 |
| Simon: Hand | 0.65 | 1.00 | 5,851.97 | 30.00 | 0.65 | 195.07 | 0.00 | 0.00 | 0.954 | 0.95 | 0.18 |
| Valence: Hand | 22.65 | 1.00 | 5,254.47 | 30.00 | 22.65 | 175.15 | 0.13 | 0.00 | 0.722 | 0.71 | 0.19 |
| Cue: Simon: Valence | 34.07 | 1.00 | 3,959.30 | 30.00 | 34.07 | 131.98 | 0.26 | 0.01 | 0.615 | 0.61 | 0.21 |
| Cue: Simon: Hand | 6.32 | 1.00 | 6,284.80 | 30.00 | 6.32 | 209.49 | 0.03 | 0.00 | 0.863 | 0.87 | 0.18 |
| Cue: Valence: Hand | **632.26** | **1.00** | **2,932.87** | **30.00** | **632.26** | **97.76** | **6.47** | **0.18** | **0.016** | **0.02** | **3.70** |
| Simon: Valence: Hand | **1,146.20** | **1.00** | **3,582.92** | **30.00** | **1,146.20** | **119.43** | **9.60** | **0.24** | **0.004** | **0.00** | **13.27** |
| Cue: Simon: Valence: Hand | 309.81 | 1.00 | 3,825.32 | 30.00 | 309.81 | 127.51 | 2.43 | 0.01 | 0.130 | 0.13 | 0.60 |

**Note:**
Bold entries indicate significance.

d. Effect refers to the difference compatible—incompatible. All $p$'s are adjusted according to *Benjamini & Yekutieli (2001)*.

# RESULTS

## Response times

We assess whether the effect of valence is rather on attentional processes or on conflict processing as both have been shown to play a role in the hand proximity effect. In other words, we assume that valence modulates (a) the interaction of cue and hand position (allocation of attention) as well as (b) the interaction of Simon and hand position (conflict processing).

Regarding the test of our first hypothesis, regarding valence affection allocation of attention (valence interacting with cue and hand position), we found a significant cue × valence × hand position interaction, $F(1,30) = 6.467$, $p_{boot} = 0.016$, $\eta_p^2 = 0.177$, $BF_{10} = 3.70$. Regarding our second hypothesis, concerning valence affecting conflict processing (valence interacting with Simon and hand position), we found a significant Simon × valence × hand position interaction, $F(1,30) = 9.597$, $p_{boot} = 0.003$, $\eta_p^2 = 0.243$, $BF_{10} = 13.266$. Overall, we found no significant four-way interaction[2] indicating that the significant Simon × cue interaction, $F(1,30) = 8.026$, $p_{boot} = 0.008$, $\eta_p^2 = 0.211$, $BF_{10} = 7.08$, is independent of both three-way interactions. Table 1 shows the complete ANOVA table.

(a) Thus, to break down the above described three-way interactions (cue × valence × hand position and Simon × Valence × hand position), we calculated follow up contrasts regarding cueing and Simon effect.

[2] With the factors gaze cueing (incongruent, congruent), Simon (compatible, incompatible) hand position (body, monitor), and valence (angry, neutral).

**Table 2 Contrasts Cue × Valence × Hand.**

| Contrast | Valence | Hand | Effect | SE | df | t | p | d | CI |
|---|---|---|---|---|---|---|---|---|---|
| Congruent—incongruent | Angry | Body | −2.44 | 2.10 | 30 | −1.16 | 0.554 | −0.21 | [−0.57 to 0.15] |
| Congruent—incongruent | Neutral | Body | −6.13 | 2.39 | 30 | −2.57 | 0.064 | −0.47 | [−0.84 to 0.09] |
| Congruent—incongruent | Angry | Monitor | −7.79 | 2.54 | 30 | −3.06 | 0.038 | −0.56 | [−0.94 to 0.17] |
| Congruent—incongruent | Neutral | Monitor | −2.45 | 2.16 | 30 | −1.13 | 0.554 | −0.21 | [−0.57 to 0.16] |

**Note:**
Note that the ANOVA has been conducted with the square-root transformed error rate.

**Table 3 Contrasts Simon × Valence × Hand.**

| Contrast | Valence | Hand | Effect | SE | df | t | p | d | CI |
|---|---|---|---|---|---|---|---|---|---|
| Compatible—incompatible | Angry | Body | −29.85 | 3.96 | 30 | −7.54 | <0.001 | −1.38 | [−1.87 to −0.87] |
| Compatible—incompatible | Neutral | Body | −22.00 | 3.79 | 30 | −5.81 | <0.001 | −1.06 | [−1.50 to −0.61] |
| Compatible—incompatible | Angry | Monitor | −23.92 | 4.01 | 30 | −5.96 | <0.001 | −1.09 | [−1.53 to −0.63] |
| Compatible—incompatible | Neutral | Monitor | −28.23 | 4.15 | 30 | −6.80 | <0.001 | −1.24 | [−1.71 to −0.76] |

**Note:**
Note that SE is the within-subjects standard error according to *Cousineau (2005, 2012)*. The confidence intervals (CI) refer to Cohen's d. Effect refers to the difference compatible—incompatible. All *p*'s are adjusted according to *Benjamini & Yekutieli (2001)*.

Table 2 provides the corresponding statistical parameters. For a hand position at the monitor, the cueing effect (congruent—incongruent) was larger for angry valences, (−7.79 ms, $SE = 2.54$, $t(30) = −3.06$, $p = 0.038$, **$d = −0.56$**), compared to neutral valence (−2.45 ms, $SE = 2.16$, $t(30) = −1.13$, $p = 0.554$, **$d = −0.21$**). This pattern was inversed for hand positions close to the body (cf. Table 2).

(b) Regarding the Simon effect (compatible-incompatible), the pattern was the other way around (cf. Table 3 for all statistical parameters): For hand positions at the monitor, the Simon effect for neutral cues was −28.23 ms, $SE = 4.15$, $t(30) = −6.80$, $p < 0.001$, **$d = −1.24$**; and thus, larger compared to angry cues, −23.92, $SE = 4.01$, $t(30) = −5.96$, $p < 0.001$, **$d = −1.09$**. Regarding a hand position close to the body, the Simon effect was larger for angry valence, −29.85 ms, $SE = 3.96$, $t(30) = −7.54$, $p < 0.001$, **$d = −1.38$**, compared to neutral valence, −22.00 ms, $SE = 3.79$, $t(30) = −5.81$, $p < 0.001$, **$d = −1.06$**.

Figure 3 provides an overview of the directions of the effects.

Regarding the main effects, we found a significant valence effect, $F(1,30) = 5.251$, $p = 0.029$, $\eta_p^2 = 0.149$, a significant cue effect, $F(1,30) = 11.111$, $p = 0.002$, $\eta_p^2 = 0.273$ and a significant Simon effect, $F(1,30) = 58.478$, $p < 0.001$, $\eta_p^2 = 0.661$. Further, there was a significant cue × Simon interaction, *i.e.*, larger Simon effect for incongruent compared to congruent cues, $F(1,30) = 8.025$, $p = 0.008$, $\eta_p^2 = 0.211$. *Post-hoc* contrasts revealed that there was a significant cueing effect (congruent-incongruent) for the incompatible Simon condition, −9.42 ms, $t(30) = −4.46$, $p < 0.001$, **$d = −0.81$**, but the cueing effect was absent in the compatible Simon condition, 0.02, $t(30) = 0.01$, $p = 1$, **$d = 0.00$**.

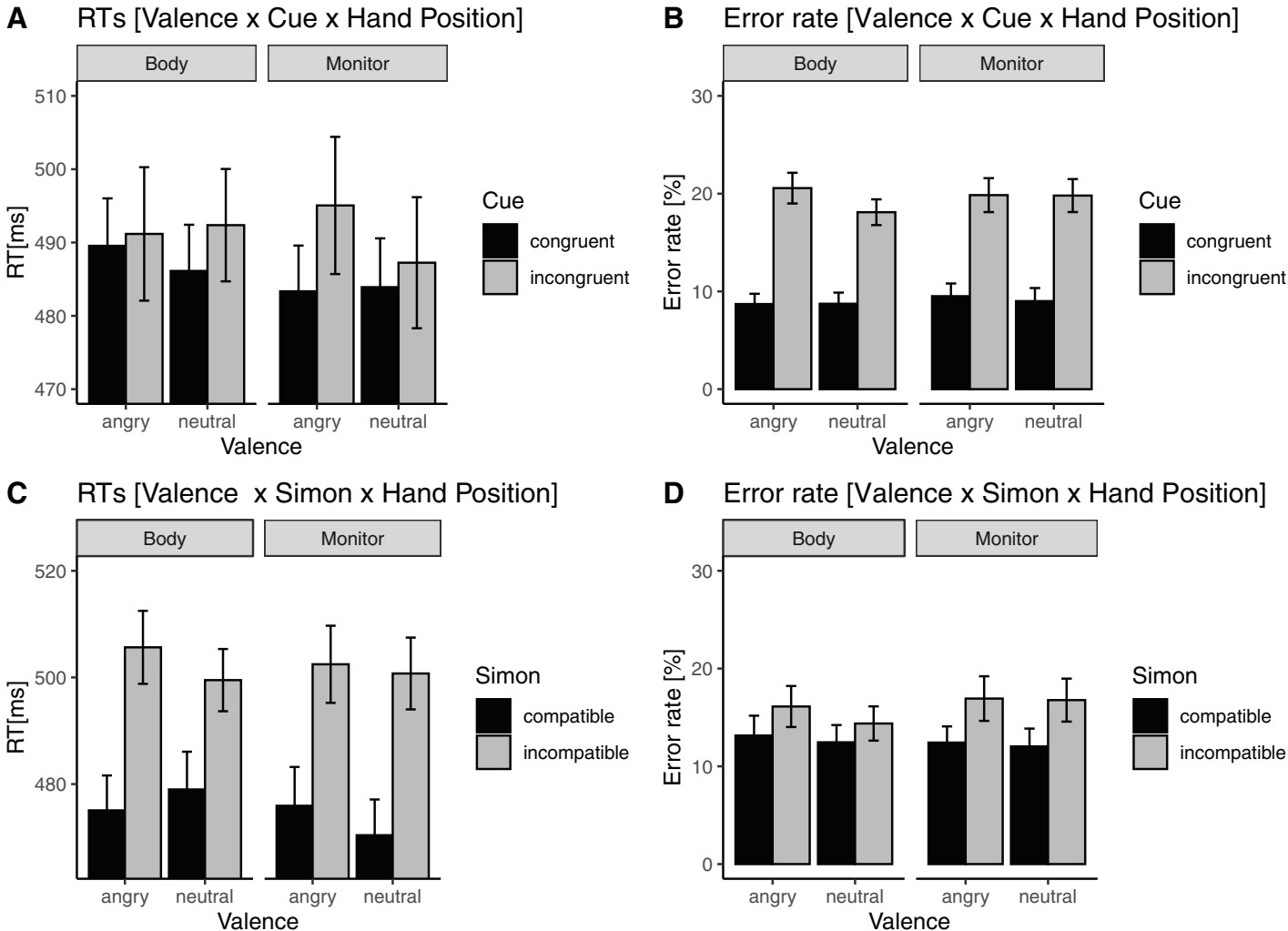

**Figure 3 Significant three-way interactions of response time.** (A) Mean response times (RTs) for the valence × cue × hand position interaction. Error bars indicate within subject 95% confidence interval according to *Cousineau (2005*, *2012)*. (C) Mean response times (RTs) for the valence × Simon effect × hand position interaction. For convenience, the corresponding error-rates (B and D) are shown, indicating that errors do not modulate the RT interactions.

## Accuracy

With respect to accuracy, we found (a) a significant cue effect, *i.e.*, higher accuracy for congruent compared to incongruent cues, $F(1,30) = 460.927$, $p_{boot} < 0.001$, $\eta_p^2 = 0.939$, $BF_{10} = 1.174101e+18$. Further, we found (b) a significant Simon effect, *i.e.*, higher accuracy for compatible compared to incompatible targets, $F(1,30) = 13.60$, $p_{boot} < 0.001$, $\eta_p^2 = 0.312$, $BF_{10} = 59.02$. In addition, there was a significant valence × cue interaction, $F(1,30) = 4.655$, $p_{boot} = 0.0396$, $\eta_p^2 = 0.134$, $BF_{10} = 1.68$, and a significant Simon × hand interaction, $F(1,30) = 13.123$, $p_{boot} = 0.001$, $\eta_p^2 = 0.304$, $BF_{10} = 49.758$. Table 4 shows the complete ANOVA table for the accuracy data.

*Post-hoc* contrasts (Table 5) revealed that the cueing effect was marginally larger for the angry valence, −1.33 ms, $SE = 0.07$, $t(30) = -19.56$, $p < 0.001$, **d = −3.57**, compared to the neutral valence, −1.18 ms, SE = 0.07, t(30) = −17.65, $p < 0.001$, **d = −3.22**.

**Table 4 Four-way ANOVA table for accuracy data.**

| Effect | SSn | dfn | SSd | dfd | MSEn | MSEd | F | p | $p_{boot}$ | $\eta_p^2$ | $BF_{10}$ |
|---|---|---|---|---|---|---|---|---|---|---|---|
| Cue | **195.53** | **1.00** | **12.73** | **30.00** | **195.53** | **0.42** | **460.93** | **<0.001** | **0.00** | **0.94** | **1.174101e+18** |
| Simon | **21.71** | **1.00** | **47.90** | **30.00** | **21.71** | **1.60** | **13.60** | **0.001** | **0.00** | **0.31** | **59.02** |
| Valence | 0.35 | 1.00 | 6.90 | 30.00 | 0.35 | 0.23 | 1.51 | 0.228 | 0.23 | 0.05 | 0.39 |
| Hand | 0.56 | 1.00 | 24.33 | 30.00 | 0.56 | 0.81 | 0.70 | 0.411 | 0.40 | 0.02 | 0.26 |
| Cue: Simon | 0.00 | 1.00 | 10.58 | 30.00 | 0.00 | 0.35 | 0.00 | 0.950 | 0.96 | 0.00 | 0.18 |
| Cue: Valence | **0.65** | **1.00** | **4.21** | **30.00** | **0.65** | **0.14** | **4.65** | **0.039** | **0.04** | **0.13** | **1.68** |
| Simon: Valence | 0.26 | 1.00 | 9.63 | 30.00 | 0.26 | 0.32 | 0.82 | 0.374 | 0.37 | 0.03 | 0.27 |
| Cue: Hand | 0.00 | 1.00 | 10.81 | 30.00 | 0.00 | 0.36 | 0.00 | 0.993 | 0.99 | 0.00 | 0.18 |
| Simon: Hand | **3.27** | **1.00** | **7.47** | **30.00** | **3.27** | **0.25** | **13.12** | **0.001** | **0.00** | **0.30** | **49.76** |
| Valence: Hand | 0.15 | 1.00 | 4.37 | 30.00 | 0.15 | 0.15 | 1.00 | 0.325 | 0.33 | 0.03 | 0.30 |
| Cue: Simon: Valence | 0.03 | 1.00 | 5.09 | 30.00 | 0.03 | 0.17 | 0.17 | 0.682 | 0.69 | 0.01 | 0.20 |
| Cue: Simon: Hand | 0.02 | 1.00 | 9.62 | 30.00 | 0.02 | 0.32 | 0.06 | 0.802 | 0.80 | 0.00 | 0.19 |
| Cue: Valence: Hand | 0.71 | 1.00 | 12.94 | 30.00 | 0.71 | 0.43 | 1.65 | 0.208 | 0.21 | 0.05 | 0.41 |
| Simon: Valence: Hand | 0.04 | 1.00 | 8.64 | 30.00 | 0.04 | 0.29 | 0.12 | 0.727 | 0.73 | 0.00 | 0.19 |
| Cue: Simon: Valence: Hand | 0.02 | 1.00 | 10.70 | 30.00 | 0.02 | 0.36 | 0.04 | 0.839 | 0.84 | 0.00 | 0.18 |

**Note:**
Note that the ANOVA has been conducted with the square-root transformed error rate. Bold entries indicate significance.

**Table 5 Contrasts (error rate) of cueing × valence.**

| Contrast | Valence | Effect | SE | df | t | p | d | CI |
|---|---|---|---|---|---|---|---|---|
| Congruent—incongruent | Angry | −1.33 | 0.07 | 30 | −19.56 | <0.001 | −3.57 | [−4.53 to −2.60] |
| Congruent—incongruent | Neutral | −1.18 | 0.07 | 30 | −17.65 | <0.001 | −3.22 | [−4.10 to −2.33] |

**Note:**
Note that the effect values refer to the square-root transformed data, as the contrasts have been calculated from those. To get an impression of the raw effects refer to Fig. 4. SE refers to within-subjects SE according to *Cousineau (2005, 2012)*. The confidence interval (95% CI) refers to Cohen's d. All p's are adjusted according to *Benjamini & Yekutieli (2001)*.

**Table 6 Contrasts (error rate) Simon × hand position.**

| Contrast | Hand | Effect | SE | df | t | p | d | CI |
|---|---|---|---|---|---|---|---|---|
| Compatible—incompatible | Body | −0.26 | 0.12 | 30 | −2.17 | 0.057 | −0.40 | [−0.77 to −0.02] |
| Compatible—incompatible | Monitor | −0.58 | 0.13 | 30 | −4.61 | <0.001 | −0.84 | [−1.25 to −0.42] |

**Note:**
Note that the effect values refer to the square-root transformed data, as the contrasts have been calculated from those. To get an impression of the raw effects refer to Fig. 4. SE refers to within-subjects SE according to *Cousineau (2005, 2012)*. The confidence interval (95% CI) refers to Cohen's d. All p's are adjusted according to *Benjamini & Yekutieli (2001)*.

Also, there was a hand position × Simon interaction, *i.e.*, bigger Simon effect for hands near to the monitor compared to hands near the body, $F(1,30) = 13.12$, $p = 0.001$, $\eta_p^2 = 0.30$, which was qualified by the corresponding *post-hoc* contrast, c.f. d's in Table 6.

Figure 4 details the interaction of the accuracy data.

## DISCUSSION

We investigated whether hand proximity modulates the Simon effect, particularly after cues that were assumed to induce a negative emotional valence. A stimulus appearing in

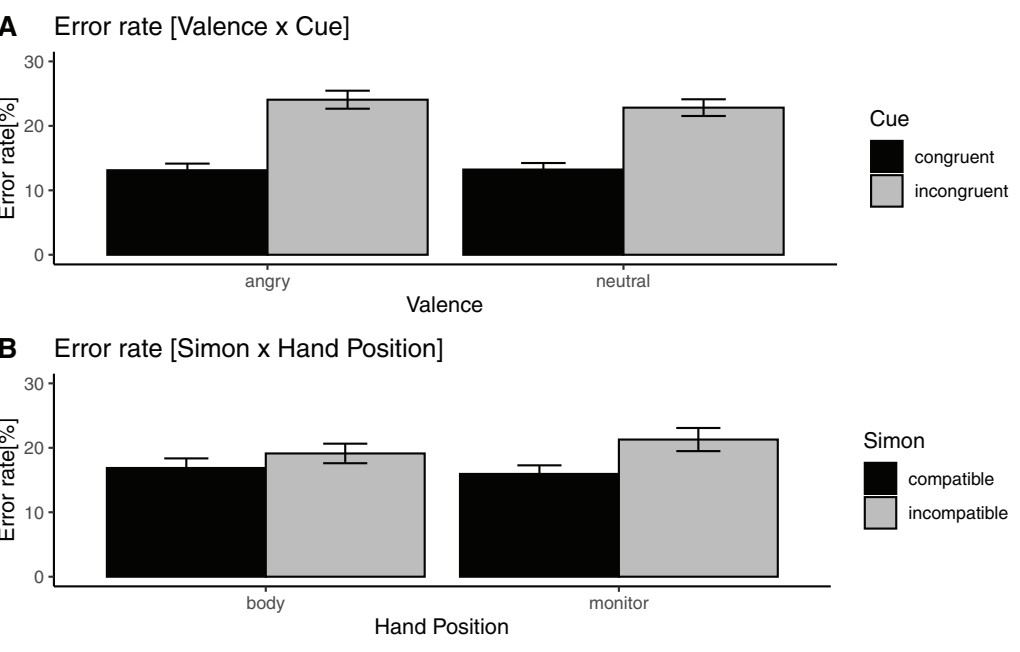

**Figure 4 Significant three-way interactions of accuracy data.** (A) Mean accuracy (%) for the valence × cue interaction. Error bars indicate within subject 95% confidence interval according to *Cousineau (2005, 2012)*. (B) Mean accuracy (%) for the Simon effect × hand position interaction.

the near hand space should become more relevant after a negative emotional cue provoking a stronger gaze cueing and thus Simon effect.

In line with our idea, that valence influences top-down allocation of attention as well as cognitive control, we found three-way interactions for reaction times: (a) A significant valence by cue by hand position interaction, *i.e.*, when hands were near the body, the cueing effect was largest with neutral incongruent cues, but if the hands were close to the monitor, the Simon effect was largest with angry incongruent cues (cf. Fig. 3). (b) Furthermore, we found a significant valence by Simon by hand position interaction. When hands were near the body, the Simon effect was largest with angry incongruent cues, but if the hands were close to the monitor, the Simon effect was largest with neutral incongruent cues (cf. Fig. 3). This interaction reflects that the direction of the Simon effect reverses for different hand positions, depending on the valence of the cues and their congruency. This indicates that hands close conditions seem to reduce the Simon effect in particular for negative stimuli. This is in line with the claim of that hands close conditions might indicate a potential for inaction in case of potentially threatening stimuli.

A somewhat unexpected result was the absent cueing effect for angry gaze cues and hands close to the body, and for neutral cues close to the monitor. Here one must consider that a hand proximity paradigm is not comparable to a standard gaze cueing paradigm. Obviously, the cueing effect is either not that stable and pronounced like in a standard manipulation, or the cueing effect is—given the order of the experimental manipulation (first instruction about hand position, then valence, followed by cueing)—more sensitive to the hand position and valence interaction. It seems like this cascade might influence cueing

in a top-down sequence. However, this should further be investigated in future experiments.

Besides the replication of previously shown hands-near effects (*Weidler & Abrams, 2014*; *Fischer & Liepelt, 2020*) in simple task variations, we extend these findings, by showing that with respect to hand position, the spatial location is not only processed with respect to cue predictability and instructional set (*Garza et al., 2013*), but there seems to be also an automatic, top-down biasing of the hand's spatial position related to the valence of the cue. But how can one interpret that the Simon effect was decreased for angry compared to neutral valences, if the hand had been close to the monitor? One explanation could be that the increased attentional effect for hands close as compared to hands far conditions increase the potential for action when the cue is neutral but decrease the potential for action when the cue has a negative valence. If the cue is negative and pointing to the incongruent side, attention remains at the cued location and the Simon effect increases. When hands are near, both, the cue and the hands may compete for attentional resources, which may explain the increase of the Simon effect in neutral *vs* angry cues. The results of the accuracy data support this interpretation: Here we found a significant valence by cue interaction such that accuracy was lower if the valence of the cue was "angry", *and* the cue was incongruent, compared to a "neutral" and incongruent cue. Further, we replicate a significant Simon by hand position effect (*Fischer & Liepelt, 2020*) for our accuracy data, such that the Simon effect was increased at the monitor as compared to the body for tasks with lower-level mappings.

Finally, when hands are spatially close to negative cues, attention might be more strongly guided towards the cued target position, and interpreted as being more meaningful, *i.e.*, predictive, compared to the neutral cues. In this condition, if the cue is negative and pointing to the incongruent side, attention directly shifts to that cued location, and facilitation with respect to the Simon effect can be observed. This result is in line with a before mentioned finding that the Simon effect is smaller at the monitor as compared to the body in a relatively difficult categorization task. This can be attributed to increased cognitive control for difficult tasks when stimuli were presented near the hands (*Liepelt & Fischer, 2016*). Thus both, task difficulty and the valence, *i.e.*, task demands, seem to be decisive for the direction and the strength of the hand proximity effects in the present study. However, the present task uses a different kind of target material as compared to *Liepelt & Fischer (2016)*, thus further studies should investigate the aspect of stimulus complexity. Nevertheless, increased cognitive control does have consequences for the gaze cueing effect. As already reported in the introduction, increased cognitive control seems not only to resolve conflict, but angry faces seem to bias cued spatial positions (*Holmes et al., 2013*). Indeed, our finding replicate behaviourally that gaze-orienting effects are modulated by valence like fear and anger, as reflected in the P1 (*Lassalle & Itier, 2013*), and that there might be an attentional bias for threatening stimuli (*Tipples, 2006*). Thus, we interpret the present results in the light of increased cognitive control that arises from interaction of cue valence and the hands near effect. It seems as if conflict is highest for "hands far and angry" and "hands near and neutral" (cf. Fig. 3).

Why could this be the case? In the hands near condition, it might be, since here hands are distant to the cue, attention is maximally guided by the valence, and thus shifts to the incongruent position. On opposite, when the hands are close to the cue (hands far from the body, but close to the monitor), attentional cueing is facilitated and thus attention shifts to the incongruent position. Now, if the hands are near a threatening stimulus, attention is bound to the hand position, leading to a decreased Simon effect. In opposite, if the cue is neutral, attention is not bound and the (cued) Simon effect is being increased. To the best of our knowledge, this is the first study showing that hands near effect and gaze cueing share functional cognitive control aspects. We assume that cognitive control is crucial in this regard.

However, there are some limitations to the study. First, it cannot be completely excluded that the valence of the stimuli is not processed in depth, meaning that the positive and neutral condition do not differ strongly, but the negative cue is kind of distractor setting the system to a more "alerted" state. Based on the design of the study we cannot exclude this interpretation. To do so, neurophysiological studies could be conducted testing the interaction between the emotional valence effect and the hand proximity effect showing if and how the proposed allocation of attention is biased and distributed between embodied and emotional systems. This comes close to ecological valid task situations, in which different systems compete for attention, but also complement each other in signalling action relevant stimuli preparing the cognitive system to do the right choice at the right moment in time. We think that this may be an interesting avenue for future research.

## ACKNOWLEDGEMENTS

We thank René Garbsch and Cecilia Diaz Luque for help with data acquisition and data preparation (thanks for the self-crafted mouse holder!) and Rabea Liebram for editing.

### Funding

This research was funded by the German Research Foundation, Grant Numbers DFG LI 2115/2-1 and LI 2115/6-1 awarded to Roman Liepelt, DFG FI 1624/5-1, awarded to Rico Fischer and HO 5054/8-1 awarded to Sven Hoffmann. The funders had no role in study design, data collection and analysis, decision to publish, or preparation of the manuscript.

### Grant Disclosures

The following grant information was disclosed by the authors:
German Research Foundation: DFG LI 2115/2-1, LI 2115/6-1, DFG FI 1624/5-1 and HO 5054/8-1.

### Competing Interests

The authors declare that they have no competing interests.

## Author Contributions

- Sven Hoffmann conceived and designed the experiments, performed the experiments, analyzed the data, prepared figures and/or tables, authored or reviewed drafts of the article, and approved the final draft.
- Rico Fischer conceived and designed the experiments, authored or reviewed drafts of the article, and approved the final draft.
- Roman Liepelt conceived and designed the experiments, authored or reviewed drafts of the article, and approved the final draft.

## Human Ethics

The following information was supplied relating to ethical approvals (*i.e.*, approving body and any reference numbers):

The present study was conducted according to the revision of the Helsinki Declaration of the World Medical Association (2013) and its ethics standards. The implementation of the study was covered by the ethics regulation regarding student psychology projects of the German Sport University Cologne. The data analysis was pseudo-anonymous coded, *i.e.*, personal data are not stored together with the participant's data (*e.g.*, names).

We collected only behavioral, *i.e.* response time data, without any cover story, or any other intervention besides simple reaction choices. Thus, the involvement of an ethical application was not necessary.

## Data Availability

The data, GNU R scripts, and Draft are available at OSF: Hoffmann, Sven, Rico Fischer, and Roman Liepelt. 2023. "Valence Moderates the Effect of Stimulus-Hand Proximity on Conflict Processing and Gaze-Cueing." OSF. April 6. osf.io/sbxc8.

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
