# Peer review of "Valence moderates the effect of stimulus-hand proximity on conflict processing and gaze-cueing"

_PeerJ, doi:10.7717/peerj.15286_

## Round 0.1 · original submission · Major Revisions

I have now received the comments from two reviewers. I want to thank both of them for their time and efforts in assessing this paper. As you will see, both reviewers provided positive and constructive comments, and I share their views. In particular, I wish to highlight some comments proposed by the two reviewers.

Reviewer 1 suggested improving your introduction by providing a more balanced discussion of the near-hand space effects, which are still debated.

Reviewer 1 also commented on your choice to use schematic faces (with low ecological validity) rather than real ones; related to this point, I also believe that Introduction would benefit if some more recent papers on gaze cueing are added; for instance, the role of affective contexts on gaze cueing has been widely discussed within recent reviews and meta-analysis (e.g. McKay et al., 2021); in addition, the use of a relatively short SOA could be justified (line 206) by referring to McKay’s work as well.

Both reviewers asked to clarify your sample size: I agree that an a priori power analysis would have been preferable; however, according to the most recent literature, performing power analyses post hoc is highly debated and interpreted as less than ideal (Dziak et al., 2020). So, clarify the rationale underlying your sample size and the stopping rule you used.

Reviewer 2 (like rev.1) suggested improving the organisation of your introduction as well; perhaps you could just reorganise it through new sub-paragraphs presenting the Stroop, gaze cueing, and valence.

References

Dziak, J. J., Dierker, L. C., & Abar, B. (2020). The interpretation of statistical power after the data have been gathered. Current Psychology, 39(3), 870–877. https://doi.org/10.1007/s12144-018-0018-1

McKay, K. T., Grainger, S. A., Coundouris, S. P., Skorich, D. P., Phillips, L. H., & Henry, J. D. (2021). Visual attentional orienting by eye gaze: A meta-analytic review of the gaze-cueing effect. Psychological Bulletin, 147(12), 1269–1289. https://doi.org/10.1037/bul0000353

Reviewer 1 ·

Basic reporting

Overall, I thought this paper was reasonably well-written and easy to follow. The article was also well-structured, all figures and tables were included, and I had ready access to the raw data. However, below, I provide some suggestions for how the authors can improve the clarity of the manuscript.

• Although the Simon effect is a fairly well-known effect in cognitive psychology, given that it is a key part of the study, I think it is important to explicitly define it where it is first mentioned (Line 71).

• My understanding is that whether or not near-hand space effects truly exist is a contentious issue. The authors do acknowledge this by making reference to studies supporting this idea (Lines 103 – 104), but to make this discussion a little more balanced, I think it would be good to expand on what these studies found and whether these findings contradict those presented earlier in the Introduction (as well as the possible reasons for why this might be the case).

• “comparable” on Line 143 – here, I wasn’t sure what the task difficulty was comparable to (e.g., do the authors just mean comparable to the difficulty of the task in previously published studies? If so, this should be made clear).

• Use of the word “latter” throughout the manuscript, sometimes it wasn’t clear what “latter” was referring to, so I would advise the authors to replace all instances of this word with what they are actually referring to.

• Figure 1 shows the target presentation duration as being 200 ms, but on Line 213, the target presentation duration is shown as being 100 ms. It is unclear which is the correct duration.

• Given that there are only two hypotheses, I think it would be more helpful for the reader to see them both restated in full at the beginning of the Results section (Lines 281 – 283) rather than having them be referred to as (a) and (b).

Experimental design

The research question was well-defined and appears to address a gap in the literature. The design of the experiment made sense in view of the research question, and it appears that the appropriate ethical guidelines were followed. Below, I provide some suggestions for where further justification/clarity would improve the manuscript, especially with respect to the sample size.

• No power analysis is presented in the paper, and instead, the sample size is determined based on the typical sample size of studies in the area (Lines 161 – 162). If possible, I would like to see the authors provide some further detail here; for instance, citations to previous studies that have used such sample sizes, as well as their observed effect sizes. Given that the authors were seeking to examine high-order interaction effects (e.g., three-way interactions), I suspect that the sample size required to achieve sufficient power is actually much higher than 32, unless near-hand space effects are particularly large. In addition, given that near-hand space effects tend to lack reliability (see, e.g., Dosso & Kingstone, 2018), I do not think it is wise to base sample-size decisions on what others have used in the past.

I think what is needed here is a more convincing argument for why the sample size of 32 is appropriate. Note that underpowered studies can lead to spurious effects emerging (i.e., Type I errors, not just Type II errors), so the presence of significant interaction effects cannot be taken as evidence of the study being appropriately powered.

• I was curious about why there were four possible target letters (Lines 209 – 211). Was this to make the task more attentionally demanding? I think a brief explanation here would be useful.

• Although the authors did observe valence effects, I found it interesting that they chose to use schematic faces (“emoticons”) rather than photos, which look more realistic. If there was a particular reason behind this decision then I think it could be articulated in the manuscript.

Validity of the findings

All screening and analysis were performed/reported rigorously, and I especially appreciated the provision of R code/Markdown files for full transparency.

The authors did a good job explaining the results in the Discussion and linking those results back to their two hypotheses.

Additional comments

Overall, I think this is a strong manuscript that addresses a gap in the literature. Where I think improvements can be made is in the justification of some methodological decisions, as well as further clarity/fuller descriptions in other sections.

·

Basic reporting

Clear and unambiguous English has been used throughout.
The authors provided references, sufficient field background/context.
The authors shared Raw data.
Th article includes all results relevant to the hypothesis.

Experimental design

Research question is relevant and meaningful.
The authors conducted this study with the prevailing ethical standards in the field.
Methods have been described with sufficient detail and information to replicate.

Validity of the findings

Findings are robust, statistically sound, and controlled.
Conclusions have been well stated.

---

## Round 0.2 · accepted · Accept

The authors have addressed all of the reviewers' comments and this paper is now ready for publication.

Reviewer 1 ·

Basic reporting

I am happy with the changes the authors have made in response to my comments and think the manuscript now reads very clearly.

Experimental design

I believe sufficient justification has now been provided about certain methodological decisions (as flagged in my previous review) and thank the authors for making those changes.

Validity of the findings

The data have been provided and rigorously analysed.

·

Basic reporting

Clear and unambiguous English has been used throughout.
The authors provided references, sufficient field background/context.
The authors shared Raw data.
Th article includes all results relevant to the hypothesis.

Experimental design

Research question is relevant and meaningful.
The authors conducted this study with the prevailing ethical standards in the field.
Methods have been described with sufficient detail and information to replicate.

Validity of the findings

Findings are robust, statistically sound, and controlled.
Conclusions have been well stated.